# Generalized Off-Policy Actor-Critic

**Shangtong Zhang, Wendelin Boehmer, Shimon Whiteson**
Department of Computer Science
University of Oxford
{shangtong.zhang, wendelin.boehmer, shimon.whiteson}@cs.ox.ac.uk

## Abstract

We propose a new objective, the *counterfactual objective*, unifying existing objectives for off-policy policy gradient algorithms in the continuing reinforcement learning (RL) setting. Compared to the commonly used *excursion objective*, which can be misleading about the performance of the target policy when deployed, our new objective better predicts such performance. We prove the Generalized Off-Policy Policy Gradient Theorem to compute the policy gradient of the counterfactual objective and use an *emphatic* approach to get an unbiased sample from this policy gradient, yielding the Generalized Off-Policy Actor-Critic (Geoff-PAC) algorithm. We demonstrate the merits of Geoff-PAC over existing algorithms in Mujoco robot simulation tasks, the first empirical success of emphatic algorithms in prevailing deep RL benchmarks.

## 1 Introduction

Reinforcement learning (RL) algorithms based on the policy gradient theorem (Sutton et al., 2000; Marbach and Tsitsiklis, 2001) have recently enjoyed great success in various domains, e.g., achieving human-level performance on Atari games (Mnih et al., 2016). The original policy gradient theorem is on-policy and used to optimize the *on-policy objective*. However, in many cases, we would prefer to learn off-policy to improve data efficiency (Lin, 1992) and exploration (Osband et al., 2018). To this end, the Off-Policy Policy Gradient (OPPG) Theorem (Degris et al., 2012; Maei, 2018; Imani et al., 2018) was developed and has been widely used (Silver et al., 2014; Lillicrap et al., 2015; Wang et al., 2016; Gu et al., 2017; Ciosek and Whiteson, 2017; Espeholt et al., 2018).

Ideally, an off-policy algorithm should optimize the off-policy analogue of the on-policy objective. In the continuing RL setting, this analogue would be the performance of the target policy in expectation w.r.t. the stationary distribution of the *target policy*, which is referred to as the *alternative life objective* (Ghiassian et al., 2018). This objective corresponds to the performance of the target policy when deployed. Previously, OPPG optimizes a different objective, the performance of the target policy in expectation w.r.t. the stationary distribution of the *behavior policy*. This objective is referred to as the *excursion objective* (Ghiassian et al., 2018), as it corresponds to the excursion setting (Sutton et al., 2016). Unfortunately, the excursion objective can be misleading about the performance of the target policy when deployed, as we illustrate in Section 3.

It is infeasible to optimize the alternative life objective directly in the off-policy continuing setting. Instead, we propose to optimize the *counterfactual objective*, which approximates the alternative life objective. In the excursion setting, an agent in the stationary distribution of the behavior policy considers a hypothetical excursion that follows the target policy. The return from this hypothetical excursion is an indicator of the performance of the target policy. The excursion objective measures this return w.r.t. the stationary distribution of the behavior policy, using samples generated by executing the behavior policy. By contrast, evaluating the alternative life objective requires samples from the stationary distribution of the target policy, to which the agent does not have access. In the counterfactual objective, we use a new parameter $\hat{\gamma}$ to control *how counterfactual the objective is*,

akin to Gelada and Bellemare (2019). With $\hat{\gamma} = 0$, the counterfactual objective uses the stationary distribution of the behavior policy to measure the performance of the target policy, recovering the excursion objective. With $\hat{\gamma} = 1$, the counterfactual objective is fully decoupled from the behavior policy and uses the stationary distribution of the target policy to measure the performance of the target policy, recovering the alternative life objective. As in the excursion objective, the excursion is never actually executed and the agent always follows the behavior policy.

We make two contributions in this paper. First, we prove the Generalized Off-Policy Policy Gradient (GOPPG) Theorem, which gives the policy gradient of the counterfactual objective. Second, using an emphatic approach (Sutton et al., 2016) to compute an unbiased sample for this policy gradient, we develop the Generalized Off-Policy Actor-Critic (Geoff-PAC) algorithm. We evaluate Geoff-PAC empirically in challenging robot simulation tasks with neural network function approximators. Geoff-PAC outperforms the actor-critic algorithms proposed by Degris et al. (2012); Imani et al. (2018), and to our best knowledge, Geoff-PAC is the first empirical success of emphatic algorithms in prevailing deep RL benchmarks.

## 2  Background

We use a time-indexed capital letter (e.g., $X_t$) to denote a random variable. We use a bold capital letter (e.g., $\mathbf{X}$) to denote a matrix and a bold lowercase letter (e.g., $\mathbf{x}$) to denote a column vector. If $x : \mathcal{S} \to \mathbb{R}$ is a scalar function defined on a finite set $\mathcal{S}$, we use its corresponding bold lowercase letter to denote its vector form, i.e., $\mathbf{x} \doteq [x(s_1), \ldots, x(s_{|\mathcal{S}|})]^{\mathrm{T}}$. We use $\mathbf{I}$ to denote the identity matrix and $\mathbf{1}$ to denote an all-one column vector.

We consider an infinite horizon MDP (Puterman, 2014) consisting of a finite state space $\mathcal{S}$, a finite action space $\mathcal{A}$, a bounded reward function $r : \mathcal{S} \times \mathcal{A} \to \mathbb{R}$ and a transition kernel $p : \mathcal{S} \times \mathcal{S} \times \mathcal{A} \to [0, 1]$. We consider a transition-based discount function (White, 2017) $\gamma : \mathcal{S} \times \mathcal{A} \times \mathcal{S} \to [0, 1]$ for unifying continuing tasks and episodic tasks. At time step $t$, an agent at state $S_t$ takes an action $A_t$ according to a policy $\pi : \mathcal{A} \times \mathcal{S} \to [0, 1]$. The agent then proceeds to a new state $S_{t+1}$ according to $p$ and gets a reward $R_{t+1}$ satisfying $\mathbb{E}[R_{t+1}] = r(S_t, A_t)$. The return of $\pi$ at time step $t$ is $G_t \doteq \sum_{i=0}^{\infty} \Gamma_t^{i-1} R_{t+1+i}$, where $\Gamma_t^{i-1} \doteq \Pi_{j=0}^{i-1} \gamma(S_{t+j}, A_{t+j}, S_{t+j+1})$ and $\Gamma_t^{-1} \doteq 1$. We use $v_\pi$ to denote the value function of $\pi$, which is defined as $v_\pi(s) \doteq \mathbb{E}_\pi[G_t | S_t = s]$. Like White (2017), we assume $v_\pi$ exists for all $s$. We use $q_\pi(s, a) \doteq \mathbb{E}_\pi[G_t | S_t = s, A_t = a]$ to denote the state-action value function of $\pi$. We use $\mathbf{P}_\pi$ to denote the transition matrix induced by $\pi$, i.e., $\mathbf{P}_\pi[s, s'] \doteq \sum_a \pi(a|s) p(s'|s, a)$. We assume the chain induced by $\pi$ is ergodic and use $\mathbf{d}_\pi$ to denote its unique stationary distribution. We define $\mathbf{D}_\pi \doteq diag(\mathbf{d}_\pi)$.

In the off-policy setting, an agent aims to learn a target policy $\pi$ but follows a behavior policy $\mu$. We use the same assumption of coverage as Sutton and Barto (2018), i.e., $\forall (s, a), \pi(a|s) > 0 \implies \mu(a|s) > 0$. We assume the chain induced by $\mu$ is ergodic and use $\mathbf{d}_\mu$ to denote its stationary distribution. Similarly, $\mathbf{D}_\mu \doteq diag(\mathbf{d}_\mu)$. We define $\rho(s, a) \doteq \frac{\pi(a|s)}{\mu(a|s)}$, $\rho_t \doteq \rho(S_t, A_t)$ and $\gamma_t \doteq \gamma(S_{t-1}, A_{t-1}, S_t)$.

Typically, there are two kinds of tasks in RL, prediction and control.

**Prediction:** In prediction, we are interested in finding the value function $v_\pi$ of a given policy $\pi$. Temporal Difference (TD) learning (Sutton, 1988) is perhaps the most powerful algorithm for prediction. TD enjoys convergence guarantee in both on- and off-policy tabular settings. TD can also be combined with linear function approximation. The update rule for on-policy linear TD is $w \leftarrow w + \alpha \Delta_t$, where $\alpha$ is a step size and $\Delta_t \doteq [R_{t+1} + \gamma V(S_{t+1}) - V(S_t)] \nabla_w V(S_t)$ is an incremental update. Here we use $V$ to denote an estimate of $v_\pi$ parameterized by $w$. Tsitsiklis and Van Roy (1997) prove the convergence of on-policy linear TD. In off-policy linear TD, the update $\Delta_t$ is weighted by $\rho_t$. The divergence of off-policy linear TD is well documented (Tsitsiklis and Van Roy, 1997). To approach this issue, Gradient TD methods (Sutton et al. 2009) were proposed. Instead of bootstrapping from the prediction of a successor state like TD, Gradient TD methods compute the gradient of the projected Bellman error directly. Gradient TD methods are true stochastic gradient methods and enjoy convergence guarantees. However, they are usually two-time-scale, involving two sets of parameters and two learning rates, which makes it hard to use in practice (Sutton et al., 2016). To approach this issue, Emphatic TD (ETD, Sutton et al. 2016) was proposed.

ETD introduces an interest function $i : \mathcal{S} \to [0, \infty)$ to specify user's preferences for different states. With function approximation, we typically cannot get accurate predictions for all states and must thus trade off between them. States are usually weighted by $d_\mu(s)$ in the off-policy setting (e.g., Gradient TD methods) but with the interest function, we can explicitly weight them by $d_\mu(s)i(s)$ in our objective. Consequently, we weight the update at time $t$ via $M_t$, which is the *emphasis* that accumulates previous interests in a certain way. In the simplest form of ETD, we have $M_t \doteq i(S_t) + \gamma_t \rho_{t-1} M_{t-1}$. The update $\Delta_t$ is weighted by $\rho_t M_t$. In practice, we usually set $i(s) \equiv 1$.

Inspired by ETD, Hallak and Mannor (2017) propose to weight $\Delta_t$ via $\rho_t \bar{c}(S_t)$ in the Consistent Off-Policy TD (COP-TD) algorithm, where $\bar{c}(s) \doteq \frac{d_\pi(s)}{d_\mu(s)}$ is the density ratio, which is also known as the covariate shift (Gelada and Bellemare, 2019). To learn $\bar{c}$ via stochastic approximation, Hallak and Mannor (2017) propose the COP operator. However, the COP operator does not have a unique fixed point. Extra normalization and projection is used to ensure convergence (Hallak and Mannor, 2017) in the tabular setting. To address this limitation, Gelada and Bellemare (2019) further propose the $\hat{\gamma}$-discounted COP operator.

Gelada and Bellemare (2019) define a new transition matrix $\mathbf{P}_{\hat{\gamma}} \doteq \hat{\gamma}\mathbf{P}_\pi + (1-\hat{\gamma})\mathbf{1}\mathbf{d}_\mu^{\mathrm{T}}$ where $\hat{\gamma} \in [0, 1]$ is a constant. Following this matrix, an agent either proceeds to the next state according to $\mathbf{P}_\pi$ w.p. $\hat{\gamma}$ or gets reset to $\mathbf{d}_\mu$ w.p. $1 - \hat{\gamma}$. Gelada and Bellemare (2019) show the chain under $\mathbf{P}_{\hat{\gamma}}$ is ergodic. With $\mathbf{d}_{\hat{\gamma}}$ denoting its stationary distribution, they prove

$$\mathbf{d}_{\hat{\gamma}} = (1 - \hat{\gamma})(\mathbf{I} - \hat{\gamma}\mathbf{P}_\pi^{\mathrm{T}})^{-1}\mathbf{d}_\mu \quad (\hat{\gamma} < 1) \tag{1}$$

and $\mathbf{d}_{\hat{\gamma}} = \mathbf{d}_\pi$ ($\hat{\gamma} = 1$). With $c(s) \doteq \frac{d_{\hat{\gamma}}(s)}{d_\mu(s)}$, Gelada and Bellemare (2019) prove that

$$\mathbf{c} = \hat{\gamma}\mathbf{D}_\mu^{-1}\mathbf{P}_\pi^{\mathrm{T}}\mathbf{D}_\mu\mathbf{c} + (1 - \hat{\gamma})\mathbf{1}, \tag{2}$$

yielding the following learning rule for estimating $c$ in the tabular setting:

$$C(S_{t+1}) \leftarrow C(S_{t+1}) + \alpha[\hat{\gamma}\rho_t C(S_t) + (1 - \hat{\gamma}) - C(S_{t+1})], \tag{3}$$

where $C$ is an estimate of $c$ and $\alpha$ is a step size. A semi-gradient is used when $C$ is a parameterized function (Gelada and Bellemare, 2019). For a small $\hat{\gamma}$ (depending on the difference between $\pi$ and $\mu$), Gelada and Bellemare (2019) prove a multi-step contraction under linear function approximation. For a large $\hat{\gamma}$ or nonlinear function approximation, they provide an extra normalization loss for the sake of the constraint $\mathbf{d}_\mu^{\mathrm{T}}\mathbf{c} = \mathbf{1}^{\mathrm{T}}\mathbf{d}_{\hat{\gamma}} = 1$. Gelada and Bellemare (2019) use $\rho_t c(S_t)$ to weight the update $\Delta_t$ in Discounted COP-TD. They demonstrate empirical success in Atari games (Bellemare et al., 2013) with pixel inputs.

**Control:** In this paper, we focus on policy-based control. In the on-policy continuing setting, we seek to optimize the average value objective (Silver, 2015)

$$J_\pi \doteq \sum_s d_\pi(s)i(s)v_\pi(s). \tag{4}$$

Optimizing the average value objective is equivalent to optimizing the average reward objective (Puterman, 2014) if both $\gamma$ and $i$ are constant (see White 2017). In general, the average value objective can be interpreted as a generalization of the average reward objective to adopt transition-based discounting and nonconstant interest function.

In the off-policy continuing setting, Imani et al. (2018) propose to optimize the excursion objective

$$J_\mu \doteq \sum_s d_\mu(s)i(s)v_\pi(s) \tag{5}$$

instead of the alternative life objective $J_\pi$. The key difference between $J_\pi$ and $J_\mu$ is how we trade off different states. With function approximation, it is usually not possible to maximize $v_\pi(s)$ for all states, which is the first trade-off we need to make. Moreover, visiting one state more implies visiting another state less, which is the second trade-off we need to make. $J_\mu$ and $J_\pi$ achieve both kinds of trade-off according to $d_\mu$ and $d_\pi$ respectively. However, it is $J_\pi$, not $J_\mu$, that correctly reflects the deploy-time performance of $\pi$, as the behavior policy will no longer matter when we deploy the off-policy learned $\pi$ in a continuing task.

In both objectives, $i(s)$ is usually set to 1. We assume $\pi$ is parameterized by $\theta$. In the rest of this paper, all gradients are taken w.r.t. $\theta$ unless otherwise specified, and we consider the gradient for only

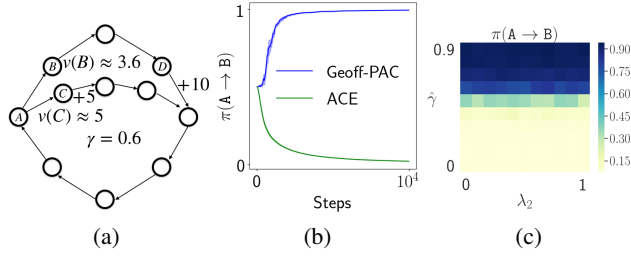

Figure 1: (a) The two-circle MDP. Rewards are 0 unless specified on the edge (b) The probability of transitioning to B from A under target policy $\pi$ during training (c) The influence of $\hat{\gamma}$ and $\lambda_2$ on the final solution found by Geoff-PAC.

one component of $\theta$ for the sake of clarity. It is not clear how to compute the policy gradient of $J_\pi$ in the off-policy continuing setting directly. For $J_\mu$, we can compute the policy gradient as

$$\nabla J_\mu = \sum_s d_\mu(s)i(s) \sum_a \big(q_\pi(s,a)\nabla \pi(a|s) + \pi(a|s)\nabla q_\pi(s,a)\big). \tag{6}$$

Degris et al. (2012) prove in the Off-Policy Policy Gradient (OPPG) theorem that we can ignore the term $\pi(s,a)\nabla q_\pi(s,a)$ without introducing bias for a tabular policy[1] when $i(s) \equiv 1$, yielding the Off-Policy Actor Critic (Off-PAC), which updates $\theta$ as

$$\theta_{t+1} = \theta_t + \alpha \rho_t q_\pi(S_t, A_t)\nabla \log \pi(A_t|S_t), \tag{7}$$

where $\alpha$ is a step size, $S_t$ is sampled from $\mathbf{d}_\mu$, and $A_t$ is sampled from $\mu(\cdot|S_t)$. For a policy using a general function approximator, Imani et al. (2018) propose a new OPPG theorem. They define

$$F_t^{(1)} \doteq i(S_t) + \gamma_t \rho_{t-1} F_{t-1}^{(1)}, \quad M_t^{(1)} \doteq (1-\lambda_1)i(S_t) + \lambda_1 F_t^{(1)},$$
$$Z_t^{(1)} \doteq \rho_t M_t^{(1)} q_\pi(S_t, A_t)\nabla \log \pi(A_t|S_t),$$

where $\lambda_1 \in [0,1]$ is a constant used to optimize the bias-variance trade-off and $F_{-1}^{(1)} \doteq 0$. Imani et al. (2018) prove that $Z_t^{(1)}$ is an unbiased sample of $\nabla J_\mu$ in the limiting sense if $\lambda_1 = 1$ and $\pi$ is fixed, i.e., $\lim_{t\to\infty} \mathbb{E}_\mu[Z_t^{(1)}] = \nabla J_\mu$. Based on this, Imani et al. (2018) propose the Actor-Critic with Emphatic weightings (ACE) algorithm, which updates $\theta$ as $\theta_{t+1} = \theta_t + \alpha Z_t^{(1)}$. ACE is an emphatic approach where $M_t^{(1)}$ is the emphasis to reweigh the update.

## 3  The Counterfactual Objective

We now introduce the counterfactual objective

$$J_{\hat{\gamma}} \doteq \sum_s d_{\hat{\gamma}}(s)\hat{i}(s)v_\pi(s), \tag{8}$$

where $\hat{i}$ is a user-defined interest function. Similarly, we can set $\hat{i}(s)$ to 1 for the continuing setting but we proceed with a general $\hat{i}$. When $\hat{\gamma} = 1$, $J_{\hat{\gamma}}$ recovers the alternative life objective $J_\pi$. When $\hat{\gamma} = 0$, $J_{\hat{\gamma}}$ recovers the excursion objective $J_\mu$. To motivate the counterfactual objective $J_{\hat{\gamma}}$, we first present the two-circle MDP (Figure 1a) to highlight the difference between $J_\pi$ and $J_\mu$.

In the two-circle MDP, an agent needs to make a decision only in state A. The behavior policy $\mu$ proceeds to B or C randomly with equal probability. For this continuing task, the discount factor $\gamma$ is always 0.6 and the interest is always 1. Under this task specification (White, 2017), the optimal policy under the alternative life objective $J_\pi$, which is equivalent to the average reward objective as $\gamma$ and $i$ are constant, is to stay in the outer circle. However, to maximize $J_\mu$, the agent prefers the inner circle. To see this, first note $v_\pi(\mathtt{B})$ and $v_\pi(\mathtt{C})$ hardly change w.r.t. $\pi$, and we have $v_\pi(\mathtt{B}) \approx 3.6$ and $v_\pi(\mathtt{C}) \approx 5$. To maximize $J_\mu$, the target policy $\pi$ would prefer transitioning to state C to maximize $v_\pi(\mathtt{A})$. The agent, therefore, remains in the inner circle. The two-circle MDP is tabular, so the policy

maximizing $v_\pi(s)$ for all $s$ can be represented accurately. If we consider an episodic task, e.g., we aim to maximize only $v_\pi(\texttt{A})$ and set the discount of the transition back to $\texttt{A}$ to 0, such policy will be optimal under the episodic return criterion. However, when we consider a continuing task and aim to optimize $J_\pi$, we have to consider state visitation. The excursion objective $J_\mu$ maximizes $v_\pi(\texttt{A})$ regardless of the state visitation under $\pi$, yielding a policy $\pi$ that never visits the state $\texttt{D}$, a state of the highest value. Such policy is sub-optimal in this continuing task. To maximize $J_\pi$, the agent has to sacrifice $v_\pi(\texttt{A})$ and visits state $\texttt{D}$ more. This two-circle MDP is not an artifact due to the small $\gamma$. The same effect can also occur with a larger $\gamma$ if the path is longer. With function approximation, the discrepancy between $J_\mu$ and $J_\pi$ can be magnified as we need to make trade-off in both maximizing $v_\pi(s)$ and state visitation.

One solution to this problem is to set the interest function $i$ in $J_\mu$ in a clever way. However, it is not clear how to achieve this without domain knowledge. Imani et al. (2018) simply set $i$ to 1. Another solution might be to optimize $J_\pi$ directly in off-policy learning, if one could use importance sampling ratios to fully correct $\mathbf{d}_\mu$ to $\mathbf{d}_\pi$ as Precup et al. (2001) propose for value-based methods in the episodic setting. However, this solution is infeasible for the continuing setting (Sutton et al., 2016). One may also use differential value function (Sutton and Barto, 2018) to replace the discounted value function in $J_\mu$. Off-policy policy gradient with differential value function, however, is still an open problem and we are not aware of existing literature on this.

In this paper, we propose to optimize $J_{\hat\gamma}$ instead. It is a well-known fact that the policy gradient of the stationary distribution exists under mild conditions (e.g., see Yu (2005)).[2] It follows immediately from the proof of this existence that $\lim_{\hat\gamma \to 1} \mathbf{d}_{\hat\gamma} = \mathbf{d}_\pi$. Moreover, it is trivial to see that $\lim_{\hat\gamma \to 0} \mathbf{d}_{\hat\gamma} = \mathbf{d}_\mu$, indicating the counterfactual objective can recover both the excursion objective and the alternative life objective smoothly. Furthermore, we show empirically that a small $\hat\gamma$ (e.g., 0.6 in the two-circle MDP and 0.2 in Mujoco tasks) is enough to generate a different solution from maximizing $J_\mu$.

## 4   Generalized Off-Policy Policy Gradient

In this section, we derive an estimator for $\nabla J_{\hat\gamma}$ and show in Proposition 1 that it is unbiased in the limiting sense. Our (standard) assumptions are given in supplementary materials. The OPPG theorem (Imani et al., 2018) leaves us the freedom to choose the interest function $i$ in $J_\mu$. In this paper, we set $i(s) \doteq \hat{i}(s)c(s)$, which, to our best knowledge, is the first time that a non-trivial interest is used. Hence, $i$ depends on $\pi$ and we cannot invoke OPPG directly as $\nabla J_\mu \neq \sum_d d_\mu(s)i(s)\nabla v_\pi(s)$. However, we can still invoke the remaining parts of OPPG:

$$\sum_s d_\mu(s)i(s)\nabla v_\pi(s) = \sum_s m(s) \sum_a q_\pi(s,a)\nabla\pi(a|s), \tag{9}$$

where $\mathbf{m}^\mathrm{T} \doteq \mathbf{i}^\mathrm{T}\mathbf{D}_\mu(\mathbf{I} - \mathbf{P}_{\pi,\gamma})^{-1}, \mathbf{P}_{\pi,\gamma}[s,s'] \doteq \sum_a \pi(a|s)p(s'|s,a)\gamma(s,a,s')$. We now compute the gradient $\nabla J_{\hat\gamma}$.

**Theorem 1 (Generalized Off-Policy Policy Gradient Theorem)**

$$\nabla J_{\hat\gamma} = \underbrace{\sum_s m(s) \sum_a q_\pi(s,a)\nabla\pi(a|s)}_{①} + \underbrace{\sum_s d_\mu(s)\hat{i}(s)v_\pi(s)g(s)}_{②} \quad (\hat\gamma < 1)$$

*where* $\mathbf{g} \doteq \hat\gamma\mathbf{D}_\mu^{-1}(\mathbf{I} - \hat\gamma\mathbf{P}_\pi^\mathrm{T})^{-1}\mathbf{b}$, $\mathbf{b} \doteq \nabla\mathbf{P}_\pi^\mathrm{T}\mathbf{D}_\mu\mathbf{c}$

*Proof.* We first use the product rule of calculus and plug in $d_{\hat\gamma}(s) = d_\mu(s)c(s)$:

$$\nabla J_{\hat\gamma} = \sum_s d_{\hat\gamma}(s)\hat{i}(s)\nabla v_\pi(s) + \sum_s \nabla d_{\hat\gamma}(s)\hat{i}(s)v_\pi(s)$$

$$= \underbrace{\sum_s d_\mu(s)c(s)\hat{i}(s)\nabla v_\pi(s)}_{③} + \underbrace{\sum_s d_\mu(s)\nabla c(s)\hat{i}(s)v_\pi(s)}_{④}.$$

$\textcircled{1} = \textcircled{3}$ follows directly from (9). To show $\textcircled{2} = \textcircled{4}$, we take gradients on both sides of (2). We have $\nabla \mathbf{c} = \hat{\gamma} \mathbf{D}_\mu^{-1} \mathbf{P}_\pi^{\mathrm{T}} \mathbf{D}_\mu \nabla \mathbf{c} + \hat{\gamma} \mathbf{D}_\mu^{-1} \nabla \mathbf{P}_\pi^{\mathrm{T}} \mathbf{D}_\mu \mathbf{c}$. Solving this linear system of $\nabla \mathbf{c}$ leads to

$$\nabla \mathbf{c} = (\mathbf{I} - \hat{\gamma} \mathbf{D}_\mu^{-1} \mathbf{P}_\pi^{\mathrm{T}} \mathbf{D}_\mu)^{-1} \hat{\gamma} \mathbf{D}_\mu^{-1} \nabla \mathbf{P}_\pi^{\mathrm{T}} \mathbf{D}_\mu \mathbf{c} = \left( \mathbf{D}_\mu^{-1} (\mathbf{I} - \hat{\gamma} \mathbf{P}_\pi^{\mathrm{T}}) \mathbf{D}_\mu \right)^{-1} \hat{\gamma} \mathbf{D}_\mu^{-1} \nabla \mathbf{P}_\pi^{\mathrm{T}} \mathbf{D}_\mu \mathbf{c}$$

$$= \left( \mathbf{D}_\mu^{-1} (\mathbf{I} - \hat{\gamma} \mathbf{P}_\pi^{\mathrm{T}})^{-1} \mathbf{D}_\mu \right) \hat{\gamma} \mathbf{D}_\mu^{-1} \nabla \mathbf{P}_\pi^{\mathrm{T}} \mathbf{D}_\mu \mathbf{c} = \mathbf{g}.$$

With $\nabla c(s) = g(s)$, $\textcircled{2} = \textcircled{4}$ follows easily. $\qquad \square$

Now we use an emphatic approach to provide an unbiased sample of $\nabla J_{\hat{\gamma}}$. We define

$$I_t \doteq c(S_{t-1}) \rho_{t-1} \nabla \log \pi(A_{t-1}|S_{t-1}), \quad F_t^{(2)} \doteq I_t + \hat{\gamma} \rho_{t-1} F_{t-1}^{(2)}, \quad M_t^{(2)} \doteq (1 - \lambda_2) I_t + \lambda_2 F_t^{(2)}.$$

Here $I_t$ functions as an intrinsic interest (in contrast to the user-defined extrinsic interest $\hat{i}$) and is a sample for $\mathbf{b}$. $F_t^{(2)}$ accumulates previous interests and translates $\mathbf{b}$ into $\mathbf{g}$. $\lambda_2$ is for bias-variance trade-off similar to Sutton et al. (2016); Imani et al. (2018). We now define

$$Z_t^{(2)} \doteq \hat{\gamma} \hat{i}(S_t) v_\pi(S_t) M_t^{(2)}, \quad Z_t \doteq Z_t^{(1)} + Z_t^{(2)}$$

and proceed to show that $Z_t$ is an unbiased sample of $\nabla J_{\hat{\gamma}}$ when $t \to \infty$.

**Lemma 1** *Assuming the chain induced by $\mu$ is ergodic and $\pi$ is fixed, the limit $f(s) \doteq d_\mu(s) \lim_{t \to \infty} \mathbb{E}_\mu[F_t^{(2)}|S_t = s]$ exists, and $\mathbf{f} = (\mathbf{I} - \hat{\gamma} \mathbf{P}_\pi^{\mathrm{T}})^{-1} \mathbf{b}$ for $\hat{\gamma} < 1$.*

*Proof.* Previous works (Sutton et al., 2016; Imani et al., 2018) assume $\lim_{t \to \infty} \mathbb{E}_\mu[F_t^{(1)}|S_t = s]$ exists. Here we prove the existence of $\lim_{t \to \infty} \mathbb{E}_\mu[F_t^{(2)}|S_t = s]$, inspired by the process of computing the value of $\lim_{t \to \infty} \mathbb{E}_\mu[F_t^{(1)}|S_t = s]$ (assuming its existence) in Sutton et al. (2016). The existence of $\lim_{t \to \infty} \mathbb{E}_\mu[F_t^{(1)}|S_t = s]$ with transition-dependent $\gamma$ can also be established with the same routine.[3] The proof also involves similar techniques as Hallak and Mannor (2017). Details in supplementary materials. $\qquad \square$

**Proposition 1** *Assuming the chain induced by $\mu$ is ergodic, $\pi$ is fixed, $\lambda_1 = \lambda_2 = 1, \hat{\gamma} < 1, i(s) \doteq \hat{i}(s)c(s)$, then $\lim_{t \to \infty} \mathbb{E}_\mu[Z_t] = \nabla J_{\hat{\gamma}}$*

*Proof.* The proof involves Proposition 1 in Imani et al. (2018) and Lemma 1. Details are provided in supplementary materials. $\qquad \square$

When $\hat{\gamma} = 0$, the Generalized Off-Policy Policy Gradient (GOPPG) Theorem recovers the OPPG theorem in Imani et al. (2018). The main contribution of GOPPG lies in the computation of $\nabla \mathbf{c}$, i.e., the policy gradient of a distribution, which has not been done by previous policy gradient methods. The main contribution of Proposition 1 is the trace $F_t^{(2)}$, which is an effective way to approximate $\nabla \mathbf{c}$. Inspired by Propostion 1, we propose to update $\theta$ as $\theta_{t+1} = \theta_t + \alpha Z_t$, where $\alpha$ is a step size.

So far, we discussed the policy gradient for a single dimension of the policy parameter $\theta$, so $F_t^{(1)}, M_t^{(1)}, F_t^{(2)}, M_t^{(2)}$ are all scalars. When we compute policy gradients for the whole $\theta$ in parallel, $F_t^{(1)}, M_t^{(1)}$ remain scalars while $F_t^{(2)}, M_t^{(2)}$ become vectors of the same size as $\theta$. This is because our intrinsic interest "function" $I_t$ is a multi-dimensional random variable, instead of a deterministic scalar function like $\hat{i}$. We, therefore, generalize the concept of interest.

So far, we also assumed access to the true density ratio $c$ and the true value function $v_\pi$. We can plug in their estimates $C$ and $V$, yielding the Generalized Off-Policy Actor-Critic (Geoff-PAC) algorithm.[4] The density ratio estimate $C$ can be learned via the learning rule in (3). The value estimate $V$ can be learned by any off-policy prediction algorithm, e.g., one-step off-policy TD (Sutton and Barto, 2018), Gradient TD methods, (Discounted) COP-TD or V-trace (Espeholt et al., 2018). Pseudocode of Geoff-PAC is provided in supplementary materials.

We now discuss two potential practical issues with Geoff-PAC. First, Proposition 1 requires $t \to \infty$. In practice, this means $\mu$ has been executed for a long time and can be satisfied by a warm-up before

training. Second, Proposition 1 provides an unbiased sample for a *fixed* policy $\pi$. Once $\pi$ is updated, $F_t^{(1)}, F_t^{(2)}$ will be invalidated as well as $C, V$. As their update rule does not have a learning rate, we cannot simply use a larger learning rate for $F_t^{(1)}, F_t^{(2)}$ as we would do for $C, V$. This issue also appeared in Imani et al. (2018). In principle, we could store previous transitions in a replay buffer (Lin, 1992) and replay them for a certain number of steps after $\pi$ is updated. In this way, we can satisfy the requirement $t \rightarrow \infty$ and get the up-to-date $F_t^{(1)}, F_t^{(2)}$. In practice, we found this unnecessary. When we use a small learning rate for $\pi$, we assume $\pi$ changes slowly and ignore this invalidation effect.

## 5 Experimental Results

Our experiments aim to answer the following questions. 1) Can Geoff-PAC find the same solution as on-policy policy gradient algorithms in the two-circle MDP as promised? 2) How does the degree of counterfactualness ($\hat{\gamma}$) influence the solution? 3) Can Geoff-PAC scale up to challenging tasks like robot simulation in Mujoco with neural network function approximators? 4) Can the counterfactual objective in Geoff-PAC translate into performance improvement over Off-PAC and ACE? 5) How does Geoff-PAC compare with other downstream applications of OPPG, e.g., DDPG (Lillicrap et al., 2015) and TD3 (Fujimoto et al., 2018)?

### 5.1 Two-circle MDP

We implemented a tabular version of ACE and Geoff-PAC for the two-circle MDP. The behavior policy $\mu$ was random, and we monitored the probability from A to B under the target policy $\pi$. In Figure 1b, we plot $\pi(\text{A} \rightarrow \text{B})$ during training. The curves are averaged over 10 runs and the shaded regions indicate standard errors. We set $\lambda_1 = \lambda_2 = 1$ so that both ACE and Geoff-PAC are unbiased. For Geoff-PAC, $\hat{\gamma}$ was set to 0.9. ACE converges to the correct policy that maximizes $J_\mu$ as expected, while Geoff-PAC converges to the policy that maximizes $J_\pi$, the policy we want in on-policy training. Figure 1c shows how manipulating $\hat{\gamma}$ and $\lambda_2$ influences the final solution. In this two-circle MDP, $\lambda_2$ has little influence on the final solution, while manipulating $\hat{\gamma}$ significantly changes the final solution.

### 5.2 Robot Simulation

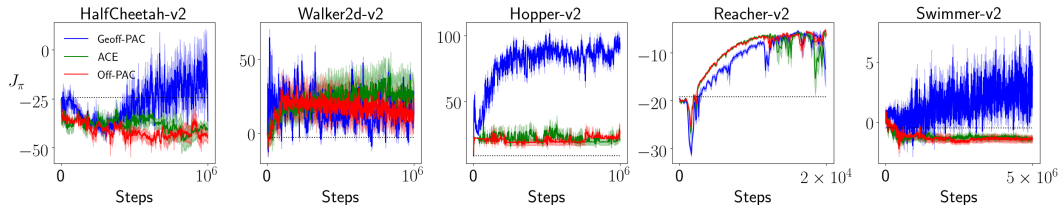

Figure 2: Comparison among Off-PAC, ACE, and Geoff-PAC. Black dash lines are random agents.

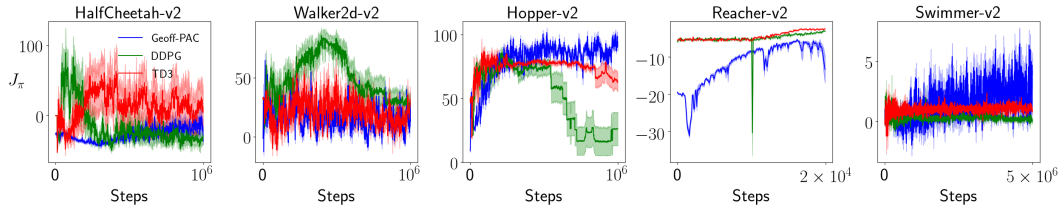

Figure 3: Comparison among DDPG, TD3, and Geoff-PAC

**Evaluation:** We benchmarked Off-PAC, ACE, DDPG, TD3, and Geoff-PAC on five Mujoco robot simulation tasks from OpenAI gym (Brockman et al., 2016). As all the original tasks are episodic, we adopted similar techniques as White (2017) to compose continuing tasks. We set the discount function $\gamma$ to 0.99 for all non-termination transitions and to 0 for all termination transitions. The

agent was teleported back to the initial states upon termination. The interest function was always 1. This setting complies with the common training scheme for Mujoco tasks (Lillicrap et al., 2015; Asadi and Williams, 2016). However, we interpret the tasks as continuing tasks. As a consequence, $J_\pi$, instead of episodic return, is the proper metric to measure the performance of a policy $\pi$. The behavior policy $\mu$ is a fixed *uniformly random policy*, same as Gelada and Bellemare (2019). The data generated by $\mu$ is significantly different from any meaningful policy in those tasks. Thus, this setting exhibits a high degree of off-policyness. We monitored $J_\pi$ periodically during training. To evaluate $J_\pi$, states were sampled according to $\pi$, and $v_\pi$ was approximated via Monte Carlo return. Evaluation based on the commonly used total undiscounted episodic return criterion is provided in supplementary materials. The relative performance under the two criterion is almost identical.

**Implementation:** Although emphatic algorithms have enjoyed great theoretical success (Yu, 2015; Hallak et al., 2016; Sutton et al., 2016; Imani et al., 2018), their empirical success is still limited to simple domains (e.g., simple hand-crafted Markov chains, cart-pole balancing) with linear function approximation. To our best knowledge, this is the first time that emphatic algorithms are evaluated in challenging robot simulation tasks with neural network function approximators. To stabilize training, we adopted the A2C (Clemente et al., 2017) paradigm with multiple workers and utilized a target network (Mnih et al., 2015) and a replay buffer (Lin, 1992). All three algorithms share the same architecture and the same parameterization. We first tuned hyperparameters for Off-PAC. ACE and Geoff-PAC inherited common hyperparameters from Off-PAC. For DDPG and TD3, we used the same architecture and hyperparameters as Lillicrap et al. (2015) and Fujimoto et al. (2018) respectively. More details are provided in supplementary materials and all the implementations are publicly available [5].

**Results:** We first studied the influence of $\lambda_1$ on ACE and the influence of $\lambda_1, \lambda_2, \hat{\gamma}$ on Geoff-PAC in HalfCheetah. The results are reported in supplementary materials. We found ACE was not sensitive to $\lambda_1$ and set $\lambda_1 = 0$ for all experiments. For Geoff-PAC, we found $\lambda_1 = 0.7, \lambda_2 = 0.6, \hat{\gamma} = 0.2$ produced good empirical results and used this combination for all remaining tasks. All curves are averaged over 10 independent runs and shaded regions indicate standard errors. Figure 2 compares Geoff-PAC, ACE, and Off-PAC. Geoff-PAC significantly outperforms ACE and Off-PAC in 3 out of 5 tasks. The performance on Walker and Reacher is similar. This performance improvement supports our claim that optimizing $J_{\hat{\gamma}}$ can better approximate $J_\pi$ than optimizing $J_\mu$. We also report the performance of a random agent for reference. Moreover, this is the first time that ACE is evaluated on such challenging domains instead of simple Markov chains. Figure 3 compares Geoff-PAC, DDPG, and TD3. Geoff-PAC outperforms DDPG in Hopper and Swimmer. DDPG with a uniformly random policy exhibits high instability in HalfCheetah, Walker, and Hopper. This is expected because DDPG ignores the discrepancy between $\mathbf{d}_\mu$ and $\mathbf{d}_\pi$. As training progresses, this discrepancy gets larger and finally yields a performance drop. TD3 uses several techniques to stabilize DDPG, which translate into the performance and stability improvement over DDPG in Figure 3. However, Geoff-PAC still outperforms TD3 in Hopper and Swimmer. This is not a fair comparison in that many design choices for DDPG, TD3 and Geoff-PAC are different (e.g., one worker vs. multiple workers, deterministic vs. stochastic policy, network architectures), and we do not expect Geoff-PAC to outperform all applications of OPPG. However, this comparison does suggest GOPPG sheds light on how to improve applications of OPPG.

## 6 Related Work

The density ratio $\mathbf{c}$ is a key component in Geoff-PAC, which is proposed by Gelada and Bellemare (2019). However, how we use this density ratio is different. Q-Learning (Watkins and Dayan, 1992; Mnih et al., 2015) is a semi-gradient method. Gelada and Bellemare (2019) use the density ratio to reweigh the Q-Learning semi-gradient update directly. The resulting algorithm still belongs to semi-gradient methods. If we would use the density ratio to reweigh the Off-PAC update (7) directly, it would just be an actor-critic analogue of the Q-Learning approach in Gelada and Bellemare (2019). This reweighed Off-PAC, however, will no longer follow the policy gradient of the objective $J_\mu$, yielding instead "policy semi-gradient". In this paper, we use the density ratio to define a new objective, the counterfactual objective, and compute the policy gradient of this new objective directly (Theorem 1). The resulting algorithm, Geoff-PAC, still belongs to policy gradient methods (in the

limiting sense). Computing the policy gradient of the counterfactual objective requires computing the policy gradient of the density ratio, which has not been explored in Gelada and Bellemare (2019).

There have been many applications of OPPG, e.g., DPG (Silver et al., 2014), DDPG, ACER (Wang et al., 2016), EPG (Ciosek and Whiteson, 2017), and IMPALA (Espeholt et al., 2018). Particularly, Gu et al. (2017) propose IPG to unify on- and off-policy policy gradients. IPG is a mix of *gradients* (i.e., a mix of $\nabla J_\mu$ and $\nabla J_\pi$). To compute $\nabla J_\pi$, IPG does need on-policy samples. In this paper, the counterfactual objective is a mix of *objectives*, and we do not need on-policy samples to compute the policy gradient of the counterfactual objective. Mixing $\nabla J_{\hat\gamma}$ and $\nabla J_\pi$ directly in IPG-style is a possibility for future work.

There have been other policy-based off-policy algorithms. Maei (2018) provide an unbiased estimator (in the limiting sense) for $\nabla J_\mu$, assuming the value function is linear. Theoretical results are provided without empirical study. Imani et al. (2018) eliminate this linear assumption and provide a thorough empirical study. We, therefore, conduct our comparison with Imani et al. (2018) instead of Maei (2018). In another line of work, the policy entropy is used for reward shaping. The target policy can then be derived from the value function directly (O'Donoghue et al., 2016; Nachum et al., 2017a; Schulman et al., 2017). This line of work includes the deep energy-based RL (Haarnoja et al., 2017, 2018), where a value function is learned off-policy and the policy is derived from the value function directly, and path consistency learning (Nachum et al., 2017a,b), where gradients are computed to satisfy certain path consistencies. This line of work is orthogonal to this paper, where we compute the policy gradients of the counterfactual objective directly in an off-policy manner and do not involve reward shaping.

Liu et al. (2018) prove that $\bar{c}$ is the unique solution for a minimax problem, which involves maximization over a function set $\mathcal{F}$. They show that theoretically $\mathcal{F}$ should be sufficiently rich (e.g., neural networks). To make it tractable, they restrict $\mathcal{F}$ to a ball of a reproducing kernel Hilbert space, yielding a closed form solution for the maximization step. SGD is then used to learn an estimate for $\bar{c}$ in the minimization step, which is then used for policy evaluation. In a concurrent work (Liu et al., 2019), this approximate for $\bar{c}$ is used in off-policy policy gradient for $J_\pi$, and empirical success is observed in simple domains. By contrast, our $J_{\hat\gamma}$ unifies $J_\pi$ and $J_\mu$, where $\hat\gamma$ naturally allows bias-variance trade-off, yielding an empirical success in challenging robot simulation tasks.

# 7   Conclusions

In this paper, we introduced the counterfactual objective unifying the excursion objective and the alternative life objective in the continuing RL setting. We further provided the Generalized Off-Policy Policy Gradient Theorem and corresponding Geoff-PAC algorithm. GOPPG is the first example that a non-trivial interest function is used, and Geoff-PAC is the first empirical success of emphatic algorithms in prevailing deep RL benchmarks. There have been numerous applications of OPPG including DDPG, ACER, IPG, EPG and IMPALA. We expect GOPPG to shed light on improving those applications. Theoretically, a convergent analysis of Geoff-PAC involving compatible function assumption (Sutton et al., 2000) or multi-timescale stochastic approximation (Borkar, 2009) is also worth further investigation.

**Acknowledgments**

SZ is generously funded by the Engineering and Physical Sciences Research Council (EPSRC). This project has received funding from the European Research Council under the European Union's Horizon 2020 research and innovation programme (grant agreement number 637713). The experiments were made possible by a generous equipment grant from NVIDIA. The authors thank Richard S. Sutton, Matthew Fellows, Huizhen Yu for the valuable discussion.

## Footnotes

[1]See Errata in Degris et al. (2012), also in Imani et al. (2018); Maei (2018).

[2] For completeness, we include that proof in supplementary materials.

[3]This existence does not follow directly from the convergence analysis of ETD in Yu (2015).

[4]At this moment, a convergence analysis of Geoff-PAC is an open problem.

[5]`https://github.com/ShangtongZhang/DeepRL`

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
