[Supplementary Material · geoff-pac-full.pdf]

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

# A Assumptions and Proofs

## A.1 Assumptions

We use the same standard assumptions as Yu (2015) and Imani et al. (2018).

## A.2 Proof of the Existence of $\nabla \mathbf{d}_{\hat{\gamma}}$

*Proof.* (From Yu (2005)) The stationary distribution $\mathbf{d}_{\hat{\gamma}}$ is an eigenvector of $\mathbf{P}_{\hat{\gamma}}$ associated with the eigenvalue 1, which is the largest eigenvalue. For any primitive matrix $\mathbf{X}$, Seneta (2006) states in the proof of his Theorem 1.1 (f) that each row of $Adj(\mathbf{I} - \mathbf{X})$ is an eigenvector associated with the largest eigenvalue of the matrix $\mathbf{X}$, where $Adj(\mathbf{X})$ is the adjugate matrix of $\mathbf{X}$. As $\mathbf{P}_{\hat{\gamma}}$ is ergodic, it is primitive. Consequently, the rows of $Adj(\mathbf{I} - \mathbf{P}_{\hat{\gamma}})$ are the eigenvectors associated with the eigenvalue 1, which is the stationary distribution $\mathbf{d}_{\hat{\gamma}}$. According to fact that each element of $Adj(\mathbf{I} - \mathbf{P}_{\hat{\gamma}})$ is a polynomial of elements in $\mathbf{P}_{\hat{\gamma}}$, $\nabla \mathbf{d}_{\hat{\gamma}}$ exists whenever $\nabla \pi$ exists. Particularly, $\mathbf{d}_{\hat{\gamma}}$ is polynomials of $\hat{\gamma}$, it follows easily that $\lim_{\hat{\gamma} \to 1} \mathbf{d}_{\hat{\gamma}} = \mathbf{d}_{\pi}$. $\qquad\square$

## A.3 Proof of Lemma 1

*Proof.* To get $F_t^{(2)}$, we start from $F_{-1}^{(2)}$ and follow $\mu$ for $t$ steps. The expectation is taken w.r.t. to this process of following $\mu$ for $t$ steps. We use $p(s|\bar{s}, k)$ to denote the probability of transitioning to a state $s$ from a state $\bar{s}$ in $k$ steps under the target policy $\pi$. We define shorthands:

$$p_t(\bar{s}, \bar{a}, s) \doteq \Pr_{\mu}(S_{t-1} = \bar{s}, A_{t-1} = \bar{a}|S_t = s), \tag{10}$$

$$f_t(s) \doteq \mathbb{E}_{\mu}[F_t^{(2)}|S_t = s], \tag{11}$$

$$i_t(s) \doteq \mathbb{E}_{\mu}[I_t|S_t = s]. \tag{12}$$

When $t \to \infty$, the agent goes to the stationary distribution $d_{\mu}$, so

$$\lim_{t \to \infty} p_t(\bar{s}, \bar{a}, s) \tag{13}$$

$$= \lim_{t \to \infty} \frac{\Pr_{\mu}(S_{t-1} = \bar{s}, A_{t-1} = \bar{a}, S_t = s)}{d_{\mu}(s)} \quad \text{(Bayes' rule)} \tag{14}$$

$$= d_{\mu}(s)^{-1} d_{\mu}(\bar{s}) \mu(\bar{a}|\bar{s}) p(s|\bar{s}, \bar{a}). \tag{15}$$

Consequently,

$$d_{\mu}(s) \rho(\bar{s}, \bar{a}) \lim_{t \to \infty} p_t(\bar{s}, \bar{a}, s) = d_{\mu}(\bar{s}) \pi(\bar{a}|\bar{s}) p(s|\bar{s}, \bar{a}). \tag{16}$$

We first compute the limit of the intrinsic interest $I_t$:

$$\lim_{t \to \infty} i_t(s) \tag{17}$$

$$= \lim_{t \to \infty} \sum_{\bar{s}, \bar{a}} \Pr_{\mu}(S_{t-1} = \bar{s}, A_{t-1} = \bar{a}|S_t = s) \mathbb{E}_{\mu}[I_t|S_{t-1} = \bar{s}, A_{t-1} = \bar{a}] \tag{18}$$

(Law of total expectation and Markov property)

$$= d_{\mu}(s)^{-1} \sum_{\bar{s}, \bar{a}} d_{\mu}(\bar{s}) \mu(\bar{a}|\bar{s}) p(s|\bar{s}, \bar{a}) c(\bar{s}) \rho(\bar{s}, \bar{a}) \nabla \log \pi(\bar{a}|\bar{s}) \quad \text{(Eq. 13 and definition of } I_t) \tag{19}$$

$$= d_{\mu}(s)^{-1} \sum_{\bar{s}, \bar{a}} d_{\mu}(\bar{s}) \pi(\bar{a}|\bar{s}) p(s|\bar{s}, \bar{a}) c(\bar{s}) \nabla \log \pi(\bar{a}|\bar{s}) \tag{20}$$

$$= d_{\mu}(s)^{-1} \sum_{\bar{s}} d_{\mu}(\bar{s}) c(\bar{s}) \sum_{\bar{a}} \nabla \pi(\bar{a}|\bar{s}) p(s|\bar{s}, \bar{a}) \tag{21}$$

$$= d_{\mu}(s)^{-1} b(s) \quad \text{(Definition of } \mathbf{b}) \tag{22}$$

We then expend $f_t$ recursively:

$$f_t(s_k)$$
$$=\mathbb{E}_\mu[I_t + \hat{\gamma}\rho_{t-1}F^{(2)}_{t-1}|S_t = s_k]$$
$$=i_t(s_k) + \hat{\gamma}\mathbb{E}_\mu[\rho_{t-1}F^{(2)}_{t-1}|S_t = s_k]$$
$$=i_t(s_k) + \hat{\gamma}\sum_{s_{k-1},a_{k-1}} p_t(s_{k-1},a_{k-1},s_k)\mathbb{E}_\mu[\rho_{t-1}F^{(2)}_{t-1}|S_{t-1} = s_{k-1}, A_{t-1} = a_{k-1}]$$
$$=i_t(s_k) + \hat{\gamma}\sum_{s_{k-1},a_{k-1}} p_t(s_{k-1},a_{k-1},s_k)\rho(s_{k-1},a_{k-1})f_{t-1}(s_{k-1})]$$
$$=i_t(s_k) + \hat{\gamma}\sum_{s_{k-1},a_{k-1}} p_t(s_{k-1},a_{k-1},s_k)\rho(s_{k-1},a_{k-1})i_{t-1}(s_{k-1})]$$
$$+ \hat{\gamma}^2 \sum_{s_{k-1},a_{k-1}} p_t(s_{k-1},a_{k-1},s_k)\rho(s_{k-1},a_{k-1}) \sum_{s_{k-2},a_{k-2}} p_{t-1}(s_{k-2},a_{k-2},s_{k-1})\rho(s_{k-2},a_{k-2})f_{t-2}(s_{k-2})$$
$$=i_t(s_k) + \hat{\gamma}\sum_{s_{k-1},a_{k-1}} p_t(s_{k-1},a_{k-1},s_k)\rho(s_{k-1},a_{k-1})i_{t-1}(s_{k-1})$$
$$+ \hat{\gamma}^2 \sum_{s_{k-1},a_{k-1}} p_t(s_{k-1},a_{k-1},s_k)\rho(s_{k-1},a_{k-1}) \sum_{s_{k-2},a_{k-2}} p_{t-1}(s_{k-2},a_{k-2},s_{k-1})\rho(s_{k-2},a_{k-2})i_{t-2}(s_{k-2})]$$
$$+ \ldots$$
$$+ \hat{\gamma}^t \sum_{s_{k-1},a_{k-1}} p_t(s_{k-1},a_{k-1},s_k)\rho(s_{k-1},a_{k-1}) \cdots \sum_{s_{k-t},a_{k-t}} p_1(s_{k-t},a_{k-t},s_{k-t+1})\rho(s_{k-t},a_{k-t})f_0(s_{k-t})$$

This means we can expand $f_t(s_k)$ into $t + 1$ terms. For each term, we multiply it by $d_\mu(s_k)$ and compute the limit of the product as:

$$\lim_{t\to\infty} d_\mu(s_k) \sum_{s_{k-1},a_{k-1}} p_t(s_{k-1},a_{k-1},s_k)\rho(s_{k-1},a_{k-1})i_{t-1}(s_{k-1}) \qquad (23)$$

$$= \sum_{s_{k-1},a_{k-1}} d_\mu(s_k)\rho(s_{k-1},a_{k-1}) \lim_{t\to\infty} p_t(s_{k-1},a_{k-1},s_k) \lim_{t\to\infty} i_{t-1}(s_{k-1}) \qquad (24)$$

$$= \sum_{s_{k-1},a_{k-1}} d_\mu(s_{k-1})\pi(a_{k-1}|s_{k-1})p(s_k|s_{k-1},a_{k-1})d_\mu(s_{k-1})^{-1}b(s_{k-1}) \quad \text{(Eq. 16 and Eq. 17)}$$

$$\qquad (25)$$

$$= \sum_{s_{k-1}} b(s_{k-1})p(s_k|s_{k-1},1) \qquad (26)$$

$$\lim_{t\to\infty} d_\mu(s_k) \sum_{s_{k-1},a_{k-1}} p_t(s_{k-1},a_{k-1},s_k)\rho(s_{k-1},a_{k-1}) \sum_{s_{k-2},a_{k-2}} p_{t-1}(s_{k-2},a_{k-2},s_{k-1})\rho(s_{k-2},a_{k-2})i_{t-2}(s_{k-2})$$

$$= \sum_{s_{k-1},a_{k-1}} d_\mu(s_k)\rho(s_{k-1},a_{k-1}) \lim_{t\to\infty} p_t(s_{k-1},a_{k-1},s_k) \lim_{t\to\infty} \sum_{s_{k-2},a_{k-2}} p_{t-1}(s_{k-2},a_{k-2},s_{k-1})\rho(s_{k-2},a_{k-2})i_{t-2}(s_{k-2})$$

$$= \sum_{s_{k-1},a_{k-1}} d_\mu(s_{k-1})\pi(a_{k-1}|s_{k-1})p(s_k|s_{k-1},a_{k-1}) \lim_{t\to\infty} \sum_{s_{k-2},a_{k-2}} p_{t-1}(s_{k-2},a_{k-2},s_{k-1})\rho(s_{k-2},a_{k-2})i_{t-2}(s_{k-2})$$

$$= \sum_{s_{k-1},a_{k-1}} \pi(a_{k-1}|s_{k-1})p(s_k|s_{k-1},a_{k-1}) \sum_{s_{k-2}} b(s_{k-2})p(s_{k-1}|s_{k-2},1) \quad \text{(Eq. 23)}$$

$$= \sum_{s_{k-2}} b(s_{k-2})p(s_k|s_{k-2},2)$$

Putting all the limits together, we have

$$
\begin{aligned}
&f(s_k)\\
=&d_\mu(s_k)\lim_{t\to\infty}f_t(s_k)\\
=&b(s_k)\\
&+\hat\gamma\sum_{s_{k-1}}b(s_{k-1})p(s_k|s_{k-1},1)\\
&+\hat\gamma^2\sum_{s_{k-2}}b(s_{k-2})p(s_k|s_{k-2},2)\\
&+\ldots
\end{aligned}
$$

In a matrix form, we have

$$\mathbf{f}=\mathbf{b}+\hat\gamma\mathbf{P}_\pi^{\mathrm{T}}\mathbf{b}+(\hat\gamma\mathbf{P}_\pi^{\mathrm{T}})^2\mathbf{b}+\ldots$$

It follows easily that $\mathbf{f}=(\mathbf{I}-\hat\gamma\mathbf{P}_\pi^{\mathrm{T}})^{-1}\mathbf{b}$ $\qquad\square$

### A.4 Proof of Proposition 1

*Proof.* From Proposition 1 in Imani et al. (2018) [6], we have

$$\lim_{t\to\infty}\mathbb{E}_\mu\Big[\rho_t M_t^{(1)}q_\pi(S_t,A_t)\nabla\log\pi(A_t|S_t)\Big]=\textcircled{1}.$$

With $\mathbf{D}_{\hat\imath}\doteq diag(\hat{\mathbf{i}})$, we have,

$$
\begin{aligned}
\lim_{t\to\infty}\mathbb{E}_\mu[\hat\gamma v_\pi(S_t)\hat\imath(S_t)M_t^{(2)}]&=\hat\gamma\lim_{t\to\infty}\mathbb{E}_\mu\Big[\mathbb{E}_\mu[v_\pi(S_t)\hat\imath(S_t)F_t^{(2)}|S_t=s]\Big]\text{ (law of total expectation)}\\
&=\hat\gamma\lim_{t\to\infty}\sum_s d_\mu(s)\mathbb{E}_\mu[v_\pi(S_t)\hat\imath(S_t)F_t^{(2)}|S_t=s]\\
&=\hat\gamma\sum_s d_\mu(s)\hat\imath(s)v_\pi(s)\lim_{t\to\infty}\mathbb{E}_\mu[F_t^{(2)}|S_t=s]\text{ (conditional independence)}\\
&=\hat\gamma\sum_s v_\pi(s)\hat\imath(s)f(s)\\
&=\hat\gamma\mathbf{v}_\pi^{\mathrm{T}}\mathbf{D}_{\hat\imath}\mathbf{f}\\
&=\hat\gamma\mathbf{v}_\pi^{\mathrm{T}}\mathbf{D}_{\hat\imath}\mathbf{D}_\mu\mathbf{D}_\mu^{-1}(\mathbf{I}-\hat\gamma\mathbf{P}_\pi^{\mathrm{T}})^{-1}\mathbf{b}\text{ (Lemma 1)}\\
&=\mathbf{v}_\pi^{\mathrm{T}}\mathbf{D}_{\hat\imath}\mathbf{D}_\mu\mathbf{g}\\
&=\sum_s d_\mu(s)\hat\imath(s)v_\pi(s)g(s)=\textcircled{2}.
\end{aligned}
$$

As $\nabla J_{\hat\gamma}=\textcircled{1}+\textcircled{2}$, we have proved $\lim_{t\to\infty}\mathbb{E}_\mu[Z_t]=\nabla J_{\hat\gamma}$. $\qquad\square$

## B Details of Experiments

### B.1 Pseudocode of Geoff-PAC

Algorithm 1 provides the pseudocode of Geoff-PAC. SNLoss refers to the soft normalization loss for $\theta_c$ in Gelada and Bellemare (2019). $\beta$ is the weight for the SNLoss.

### B.2 Implementation Details

**Task Selection:** We use 5 Mujoco tasks from Open AI gym [7](Brockman et al., 2016). Those 5 tasks are of a medium difficulty level. The easy tasks (e.g., the pendulum tasks) and the hard tasks (e.g., the ant task and the humanoid tasks) are excluded.

**Algorithm 1:** Geoff-PAC with function approximation

---

**Input:**
$V$: value function parameterized by $\theta_v$
$C$: density ratio estimation parameterized by $\theta_c$
$\pi$: policy function parameterized by $\theta$
$\hat{i}$ : an interest function

Initialize target networks $\theta_v^- \leftarrow \theta_v, \theta_c^- \leftarrow \theta_c$
Initialize $F^{(1)} \leftarrow 0, F^{(2)} \leftarrow 0, t \leftarrow 0$
**while** *True* **do**
    Sample a transition $S_t, A_t, R_{t+1}, S_{t+1}$ according to behavior policy $\mu$
    **if** $t = 0$ **then**
        $t \leftarrow t + 1$
        continue
    **end**
    $\delta_t = R_{t+1} + \gamma_t V(S_{t+1}; \theta_v^-) - V(S_t; \theta_v)$
    Update $\theta_v$ to minimize $\rho_t \delta_t^2$
    Update $\theta_c$ to minimize $\left( \left( \hat{\gamma} \rho_t C(S_t; \theta_c^-) + (1 - \hat{\gamma}) - C(S_{t+1}; \theta_c) \right)^2 + \beta \texttt{SNLoss}(\theta_c) \right)$
    $F^{(1)} \leftarrow \gamma \rho_{t-1} F^{(1)} + \hat{i}(S_t) C(S_t; \theta_c)$
    $M^{(1)} \leftarrow (1 - \lambda_1) \hat{i}(S_t) C(S_t; \theta_c) + \lambda_1 F^{(1)}$
    $I \leftarrow C(S_{t-1}; \theta_c) \rho_{t-1} \nabla \log \pi(A_{t-1} | S_{t-1}; \theta)$
    $F^{(2)} \leftarrow \hat{\gamma} \rho_{t-1} F^{(2)} + I$
    $M^{(2)} \leftarrow (1 - \lambda_2) I + \lambda_2 F^{(2)}$
    Update $\theta$ in the direction of $\hat{\gamma} \hat{i}(S_t) V(S_t; \theta_v) M^{(2)} + \rho_t M^{(1)} \delta_t \nabla \log \pi(A_t | S_t; \theta)$
    Synchronize $\theta_v^-, \theta_c^-$ with $\theta_v, \theta_c$ periodically
    $t \leftarrow t + 1$
**end**

---

**Function Parameterization:** For Off-PAC, ACE, and Geoff-PAC, we use separate two-hidden-layer networks to parameterize $C, V$ and $\pi$. Each hidden layer has 64 hidden units and a ReLU (Nair and Hinton, 2010) activation function. Particularly, we parameterized $\pi$ as a diagonal Gaussian distribution with the mean being the output of the network. The standard derivation is a global state-independent variable. This is a common policy parameterization for continuous-action problems (Schulman et al., 2015, 2017b). For DDPG and TD3, we use the same parameterization as Lillicrap et al. (2015) and Fujimoto et al. (2018) respectively.

**Hyperparameter Tuning:** Our implementation is based on the A2C (Clemente et al., 2017) architecture. We first tune hyperparameters for Off-PAC based on the A2C implementation from Dhariwal et al. (2017) and previous experiences. Our ACE and Geoff-PAC implementations inherited common hyperparameters from the Off-PAC implementation without further fine-tuning. Previously, Off-PAC and ACE were evaluated on only simple domains with linear function approximation. To our best knowledge, we are the first to demonstrate an empirical success for them in challenging robot simulation tasks.

**Hyperparameters of Off-PAC:**
Number of workers: 10
Optimizer: RMSProp with an initial learning rate $10^{-3}$
Gradient clip by norm: 0.5
Replay buffer size: $10^6$
Batch size of the replay buffer: 10
Warm-up steps before learning: 100 environment steps
Target network update frequency: 200 optimization steps
Importance sampling ratio clip: [0, 2]

**Additional Hyperparameters of ACE:**
$\lambda_1 : 0$, tuned over $\{0, 0.1, 0.2, \ldots, 0.9, 1\}$ on HalfCheetah with a grid search

**Additional Hyperparameters of Geoff-PAC:**
Density ratio ($C$) clip: $[0, 2]$
`SNLoss` weight ($\beta$): $10^{-3}$, suggested by Gelada and Bellemare (2019)
$\lambda_1 : 0.7$, tuned over $\{0, 0.1, 0.2, \ldots, 0.9, 1\}$ on HalfCheetah
$\lambda_2 : 0.6$, tuned over $\{0, 0.1, 0.2, \ldots, 0.9, 1\}$ on HalfCheetah
$\hat{\gamma} : 0.2$, tuned over $\{0, 0.1, 0.2, \ldots, 0.9\}$ on HalfCheetah
We first select a reasonable $\hat{\gamma}$ based on some preliminary experiments. Then $\lambda_1$ and $\lambda_2$ are tuned with a grid search.

**Hyperparameters of DDPG:**
We implemented DDPG ourself. With the normal behavior policy, our implementation matched the reported performance in the literature, e.g., in Fujimoto et al. (2018). We use the same hyperparameters as Lillicrap et al. (2015). We do not use batch normalization.

**Hyperparameters of TD3:**
We implemented TD3 ourself. With the normal behavior policy, our implementation matched the reported performance in Fujimoto et al. (2018). With a uniformly random behavior policy, our implementation outperformed the implementation from Fujimoto et al. (2018) by a large margin, we therefore use our implementation for comparison. We use the same hyperparameters as Fujimoto et al. (2018).

**Computing Infrastructure:**
We conducted our experiments on an Nvidia DGX-1 with PyTorch.

### B.3 Other Experimental Results

We include a comparison under the total undiscounted episodic return criterion for reference. The results are reported in Figure 4 and Figure 5. All curves are averaged over 10 independent runs, and standard errors are reported as the shadow.

Figure 4: Comparison among Off-PAC, ACE, and Geoff-PAC. Black dash line is a random agent.

Figure 5: Comparison among DDPG, TD3, and Geoff-PAC

We studied the influence of $(\lambda_1, \lambda_2, \hat{\gamma})$ on ACE and Geoff-PAC in HalfCheetah. Results are reported in Figure 6. Five random seeds were used.

Figure 6: Hyper-parameter study on HalfCheetah (a) The influence of $\lambda_1$ on ACE (b) The influence of $\hat{\gamma}$ on Geoff-PAC (c) The influence of $\lambda_1, \lambda_2$ on Geoff-PAC