[Reviews · NeurIPS 2019]

Reviewer 1



This paper is largely well presented and is original work on an important problem. It contributes a new actor-critic approach to dealing with covariate shift in an off-policy objective. Empirically, this seems to result in some gains in performance, although it would need to be evaluated in a wider range of tasks and situations to understand if it more broadly applicable.

Reviewer 2



Originality: - Paper is the first to propose the counterfactual objective, which allows for algorithm users to smoothly interpolate between the alternative life objective and excursion objective. - The paper takes the gradient of this objective directly and find a set of updates using the same inspiration as previous methods in order to introduce a new update algorithm according to this objective. Quality: - Mathematical arguments are convincing and reasonable. - Paper is clear about not necessarily doing better than OPPG algorithms like DDPG, but instead showing the effectiveness of emphatic algorithms in a difficult domain using nonlinear function approximators. - Potential issues about the instability of estimating F, C, and V are mentioned, which could lead to the algorithm not converging as well as in theory. - Unclear how the research into GOPPG can improve applications of OPPG. Since this paper does not lead to significant results, perhaps a longer analysis of this and suggestions about improving OPPG methods would benefit the work. Significance: - First emphatic algorithm to make significant progress on the Mujoco domain. - Results are not that signifcant over SotA for these environments, and do not compare to more recent OPPG algorithms like SAC and TD3. Clarity: Paper is fairly clear and well organized. Math is understandable, and the more space taking proofs are in the appendix. Paper is clear about its downsides. Nits: L111 w.p. => with probability, w.p. is not a common acronym

Reviewer 3



Originality: My main concern is about the novelty of proposed method. Actually, the key part in the proposed counterfactual objective and the gradient is the covariate shift term d_gamma. However the way of correcting for the covariate shift are directly from [Gelada and Bellemare, 2019], and the rest is just put it into the policy gradient theorem in the ACE paper. Clarity: The paper is written in a clear way. Easy to follow. Significance: The comparison between DDPG and Geoff-PAC shows good practical performance and potential of the proposed method. I have another minor concern with respect to the implementation of the method. It looks not clear to me how the term g(s) in Thm 2 can be computed efficiently, especially it is later mentioned that the algorithm use a replay buffer. === After Rebuttal === After read authors' response about the novelty, I would like to increase my score accordingly. I hope the original contribution of this paper comparing with Gelada's work will be presented more clearly in the final version. Particularly on: 1) Explain the necessity of using a new counterfactual objective, and the advantage comparing with the direct gradient re-weighting method. 2) Highlight the novel part is how to computing F_t^{(2)}.

Reviewer 4



Originality: I think the method is new and the counterfactual objective is first studied. The objective is related to [Imani, E., Graves, E., and White, M. (2018)] and use [Gelada, C. and Bellemare, M. G. (2019)] methods to tackle the new objective. Quality: I would leave my main concern for the intuition for your objective function, and the difference between equation (4) and equation (5). Without emphatic term i(s), the main difference between the two objectives is the sample distribution. The original objective for on-policy setting is to let the distribution as $d_0(s)$, the intial distribution for $s_0$. The excursion objective can be explained as we first execute behavior policy $\mu$ until converge and then we execute target policy $\pi$ (since we are now using value function $v$ under policy $\pi$). The excursion objective make sense if $P^{\pi}$ is ergodic where we don't need to care about the initial distribution. In this case, objective (5) is almost the same except that we first run enough step to reach stationary distribution for policy $\pi$ and then we execute $\pi$ again. If (4) and (5) are not different under this reason, the counterfactual objective in this paper, which can be thought as a shrinkage version between (4) and (5), should not be much difference from the excursion objective which is already well-studied in [Imani, E., Graves, E., and White, M. (2018)]. I would need more explanation from authors for this main concern. For experimental part, figures 2,3 don't seem to show convincing result that Geoff-PAC over OPPG and DDPG. And from figure 4,5 it seems all the methods is not good enough compared to other paper's results (e.g. this paper https://arxiv.org/pdf/1812.02900.pdf). Clarity: I think the paper is easy to follow and well-organized. Significance: As I mentioned in quality part, and empirical results seem not as good as other off-policy optimization paper. And I would also like to hear more discussion on the choice of $\hat{\gamma}$, I believe we can choose that in a clever way using shrinkage method and may discuss that on the paper as well. In sum I think this is an interesting paper. But since I have a main concern on the objective the authors propose I tend to reject the paper at this moment.

[Author Response · NeurIPS 2019]

We thank all reviewers for their time and comments. Here are some general responses followed by individual ones.

**Section A: Related Work.** In response to **Reviewer 4**'s interpretation, we'll first contrast our work with Gelada's
work and ACE (Imani et al 2019), which will be included in the final version of the paper. Our work relates to Gelada's
work by borrowing their covariate shift ($c_{\hat\gamma}$). However, how we use $c_{\hat\gamma}$ is different. Q-learning is a semi-gradient method
and they reweigh the semi-gradient update with $c_{\hat\gamma}$ directly. If we would similarly reweigh the policy gradient update in
ACE, it would just be an actor-critic analogue of Gelada's Q-learning approach as **Reviewer 1** suggested. However, this
reweighed ACE will no longer follow the policy gradient of objective $J_\mu$, yielding instead a "policy semi-gradient". In
our work, we define a new objective with $c_{\hat\gamma}$ and derive policy gradients for this new objective. The resulting algorithm
still belongs to policy gradient methods. However, we then need to deal with $\nabla c_{\hat\gamma}$, i.e., compute the policy gradient
of a *distribution*. This has not been done in RL and cannot be handled by ACE. ACE is only a special case of our
work with $\hat\gamma = 0$ where $\nabla c_{\hat\gamma}$ disappears. In the on-policy setting, we do not need such gradients due to some algebraic
manipulation, which does not work for the off-policy setting. Therefore, ACE uses the sampling distribution $d_\mu$ instead
of an on-policy distribution to get around this issue. To the best of our knowledge, we are the first to address this issue
(computing policy gradients of a distribution) directly with a novel emphatic trace ($F_t^{(2)}$ in our paper). Furthermore,
our experiments are much more involved than ACE: Imani et al. evaluated ACE on several handcrafted simple MDPs
with linear function approximation, while we scale up both ACE and GeoffPAC to Mujoco with networks.

**Section B: TD3.** We will include a comparison with TD3 in the next version of the paper as shown by Figure 1.
Somewhat surprisingly, TD3 does not work better than DDPG in our setup. We took the TD3 implementation directly
from the author's GitHub and report the evaluation performance of the target policy. Using the author's original
parameters, in particular an initialization with $10^4$ random actions, we reproduced the reported results. However, in our
setup all $10^6$ samples are drawn from the random sampling policy $\mu$, and in this setting TD3 fails dramatically. This
may indicate that TD3 overfits to the common DDPG training setup and emphasizes the difficulty of our experimental
setting due to the high degree of off-policy samples.

**Section C: Objectives.** In contrast to **Reviewer 5**'s interpretation that $J_\pi$ is similar to $J_\mu$, we will clarify that for
off-policy training, the execution of $\pi$ is imaginary in both objectives. After we run $\mu$ till the chain mixes, we continue
to run $\mu$, during which time we evaluate $v_\pi(s)$ (using off-policy methods) with states sampled from $d_\mu$. The policy $\pi$ is
therefore never directly executed. Due to function approximation, we cannot maximize $v_\pi(s)$ for all states and have to
trade off. $J_\mu$ prefers to maximize $v_\pi(s)$ for those states that are often visited by $\mu$, while $J_\pi$ prefers states that are often
visited by $\pi$, same as what we prefer in on-policy continuing setting. As state visitation under $\mu$ and $\pi$ can be arbitrarily
different, so does $J_\mu$ and $J_\pi$.

**Reviewer 1:** **(i)** See Section A. **(ii)** Like Gelada and Bellemare (2019), we use a uniformly random behavior policy
to emphasize the importance for correcting the discrepancy between $d_\pi$ and $d_\mu$. When $\mu$ is changed, we may need to
change $\hat\gamma$ adaptively according to the similarity between $\pi$ and $\mu$, which we shall investigate in future work.

**Reviewer 3:** The comparison with TD3 in Section B reveals how much modern OPPG algorithms rely on sufficiently
recent on-policy sampling. When the difference between $\mu$ and $\pi$ is large, we would therefore expect GOPPG to improve
OPPG algorithms. It would also be possible to include other OPPG improvements into Geoff-PAC, e.g., a V-trace critic
or LSTM networks from IMPALA. Additionally, as DDPG often outperforms OffPAC, we would expect a deterministic
GeoffPAC to outperform vanilla GeoffPAC as well. We'll connect GOPPG and OPPG more explicitly and investigate
GeoffPAC and DDPG with the same architecture and computation resources in the final version of the paper.

**Reviewer 4: (Originality)** See Section A. **(Computation)** We use a novel emphatic trace ($F_t^{(2)}$ in L194) to estimate
$g(s)$ incrementally, which is theoretically supported by Proposition 1. Because we store trajectories in our replay buffer,
the sampled data from the buffer can be easily used to compute this trace. We'll clarify this in the final version.

**Reviewer 5: (Objectives)** See Section C. Furthermore, $\mu$ and $\pi$ are not transient policies before the MDP gets steady.
They assign different weights to different states and are never forgotten even after the MDP converges. We will clarify
this explicitly in the final version. **(Performance)** The cited paper shows indeed better performance but with a trained
expert as the behavior policy. This setup is much easier than sampling from a uniformly random policy $\mu$. As our above
comparison with TD3 in Section B demonstrates, off-policy methods are extremely sensitive to the behavior policy, and
the two setups are therefore not directly comparable.

Figure 1: A comparison with TD3. We only run TD3 for $10^6$ steps in Swimmer due to time limit. Curves are averaged
over 10 random seeds and shadowed regions indicate standard errors. Dashed line is a random policy.

[Meta-Review · NeurIPS 2019]

The paper studies off-policy actor-critic, where the objective function depends on the on-policy distribution of the target policy (called counterfactual objective in this work). It is shown that the use of target on-policy distribution can be beneficial, which is then used to derive gradient to result in the generalized actor-critic algorithm. The approach is evaluated on a few OpenAI benchmarks, comparing favorably over two off-policy actor-critic baselines. The novelty is somewhat limited, but the objective and algorithm are new. Overall, the reviewers feel the paper makes some interesting contributions. Minor/detailed comments: * Line 105: I am not sure if it is standard to refer c (the density ratio) as covariate shift. The latter (in my opinion) refers to the discrepancy between training/behavior and testing/target distributions, namely, the scenarios where c \ne 1; it does not refer to c itself. * The paper claims three contributions. However, the first (Line 45) is a bit over-claimed. The potential mismatch between target and excursion distribution (and the resulting performance degradation) is known.